# Naturally Occurring Nervonic Acid Ester Improves Myelin Synthesis by Human Oligodendrocytes

**DOI:** 10.3390/cells8080786

**Published:** 2019-07-29

**Authors:** Natalia Lewkowicz, Paweł Piątek, Magdalena Namiecińska, Małgorzata Domowicz, Radosław Bonikowski, Janusz Szemraj, Patrycja Przygodzka, Mariusz Stasiołek, Przemysław Lewkowicz

**Affiliations:** 1Department of Periodontology and Oral Diseases, Medical University of Lodz, 92-213 Lodz, Poland; 2Department of Neurology, Laboratory of Neuroimmunology, Medical University of Lodz, Pomorska Str. 251, 92-213 Lodz, Poland; 3Faculty of Biotechnology and Food Science, Lodz University of Technology, 90-924 Lodz, Poland; 4Department of Medical Biochemistry, Medical University of Lodz, 92-215 Lodz, Poland; 5Institute of Medical Biology, Polish Academy of Sciences, 93-232 Lodz, Poland

**Keywords:** oligodendrocytes, fish oil, nervonic acid, remyelination

## Abstract

The dysfunction of oligodendrocytes (OLs) is regarded as one of the major causes of inefficient remyelination in multiple sclerosis, resulting gradually in disease progression. Oligodendrocytes are derived from oligodendrocyte progenitor cells (OPCs), which populate the adult central nervous system, but their physiological capability to myelin synthesis is limited. The low intake of essential lipids for sphingomyelin synthesis in the human diet may account for increased demyelination and the reduced efficiency of the remyelination process. In our study on lipid profiling in an experimental autoimmune encephalomyelitis brain, we revealed that during acute inflammation, nervonic acid synthesis is silenced, which is the effect of shifting the lipid metabolism pathway of common substrates into proinflammatory arachidonic acid production. In the experiments on the human model of maturating oligodendrocyte precursor cells (hOPCs) in vitro, we demonstrated that fish oil mixture (FOM) affected the function of hOPCs, resulting in the improved synthesis of myelin basic protein, myelin oligodendrocyte glycoprotein, and proteolipid protein, as well as sphingomyelin. Additionally, FOM reduces proinflammatory cytokines and chemokines, and enhances fibroblast growth factor 2 (FGF2) and vascular endothelial growth factor (VEGF) synthesis by hOPCs was also demonstrated. Based on these observations, we propose that the intake of FOM rich in the nervonic acid ester may improve OL function, affecting OPC maturation and limiting inflammation.

## 1. Introduction

The dysfunction of oligodendrocytes (OLs), the cells responsible for neuron myelin sheath formation, is suggested as one of the most important factors underlying the incomplete remyelination in multiple sclerosis (MS) pathology. Myelin sheaths are built of multilamellar, spirally wrapping extensions of the OL plasma membrane. Myelin composition is characterized by remarkably high lipid content, which accounts for 70–75% of myelin dry weight, with myelin-specific proteins as the next major component. This specific myelin lipid composition distinguishes it from other cellular membranes [1]. In adults, new appearing OLs polarize from oligodendrocyte precursor cells (OPCs), which constitute approximately 6% of the total cell number in the central nervous system (CNS). The new myelin sheaths formed in remyelinating lesions are thinner than those observed during development [2], which can be related to the lack of essential lipids in adult human diet [3] in comparison to human milk [4]. The deficiency in myelin synthesis in MS is suggested to be associated with the early stages of OPC differentiation, during migration to the inflammatory site. Recent studies demonstrated that approximately 70% of lesions contained progenitors or premyelinating OLs characterized by an inability to differentiate into mature cells [5,6].

This idea is reflected by numerous studies carried out at different levels: histological, using cellular or animal models [7,8,9], as well as by epidemiological and case control studies [10,11,12]. The majority of reports described a decreased MS risk associated with the consumption of low-calorie diets rich in fish, vegetables, and plant fibers. On the contrary, a typical “Western Diet” that is rich in cow’s milk, meat, and animal fat correlated with higher MS prevalence [10]. More recent studies showed the beneficial effects of an increased intake of n-3 polyunsaturated fatty acids (PUFAs), eicosapentaenoic acid (EPA), and docosahexaenoic acid (DHA) in a variety of neurodegenerative and neurological diseases due to their neuroprotective and anti-inflammatory properties [13], whereas n-6 PUFAs were found to promote a proinflammatory environment [14]. A recent case control study that estimated PUFA intake from the overall diet reported an inverse association between MS and marine, but not for plant-derived long-chain n-3 PUFAs [15]. Moreover, it was also demonstrated that *fat-1* mice that express an n-3 desaturase, which allows them to convert n-6 PUFAs into n-3 PUFAs that were characterized by increased remyelination following the toxic injury of CNS in comparison to wild-type mice [16]. Neural cells are rich in long-chain PUFAs, with DHA (C22:6n-3) being the most abundant PUFA in neuronal phospholipids [17]. An adequate level of DHA in neural cell membranes is crucial for their functions, and prevents neuronal damage or apoptosis [18,19]. DHA also promotes progenitor cell differentiation into neural cells [20]. The low endogenous synthesis of n-3 PUFAs in the brain suggests an uptake from the dietary sources [21]. Moreover, EPA and DHA were demonstrated to cross the blood–brain barrier (BBB) by simple diffusion, mediating neuroprotection through prolonging the lifespan of glia cells, and inhibiting microglia and inflammatory cells [22].

Another molecule essential for the growth and maintenance of brain physiology and peripheral nervous tissue is nervonic acid (NA) (C24:1n-9), which bonds to sphingosine, forming sphingomyelin, a necessary component of myelin [23]. Clinical studies as well as an animal model of disease revealed that on the one hand, NA can be a marker of neurodegeneration; on the other, its intake can improve brain development [24,25]. The crucial role of NA observed in this phenomena was further supported in the study with the mothers’ milk of premature infants containing sevenfold higher concentrations of NA than milk produced after term delivery [26].

In the present study, we explored the effect of fish oil mixture (FOM) rich in NA and n-3 PUFAs on a human model of maturating OPCs. We demonstrated that FOM supplementation at the early stage of OPC maturation in vitro promotes myelin protein synthesis: myelin basic protein (MBP), myelin oligodendrocyte glycoprotein (MOG), and proteolipid protein (PLP) and sphingolipids by mature OLs, but also reduces proinflammatory cytokines and chemokines, and enhances fibroblast growth factor 2 (FGF2) and vascular endothelial growth factor (VEGF) synthesis by hOPCs.

## 2. Materials and Methods

### 2.1. Experimental Autoimmune Encephalomyelitis (EAE) Mouse Model

C57BL/6 mice were housed and maintained in an accredited facility, the Animal Core Department of the Medical University of Lodz. All the experiments were approved by the University Ethics Committee (12/ŁB702/2014). In each EAE experiment, six to eight-week-old female C57BL/6 mice were injected with MOG_35-55_ in CFA subcutaneously. On day 0, each mouse received 0.25 mL of a 0.15-mg mixture of dissolved MOG_35-55_ in 0.1 mL of PBS and 0.75 mg of Mycobacterium tuberculosis in 0.15 mL of complete Freund’s adjuvant (CFA), which was injected into four abdominal sites. The BBB was damaged by using 0.2 μg of Pertussis toxin (Sigma-Aldrich) injected into a tail vein on days 0 and 2. Mice were observed for neurological signs of EAE and were scored using the scale 0 to 5 as follows: 0—non-disease; 1—weak tail or wobbly walk; 2—hind limbs paralysis; 3—forelimbs paralysis; 4—hind and forelimb paralysis; 5—death or euthanasia for ethical reasons. Three weeks after the peak (remyelination period), mice were sacrificed and sampled for further analysis. At the peak of the disease, representative animals were perfused with 25 mL of cold-buffered 2.5% glutaraldehyde and ice-cold PBS (10 mL/min). Brains were removed, fixed in 4% paraformaldehyde (PFA), and lyophilized (all reagents from Sigma Aldrich).

### 2.2. Human Cell Model of Progenitor Cell Differentiation to Mature Myelin-Producing Oligodendrocytes

A human oligodendroglia cell line MO3.13 (OLs, Tebu-bio, Le Perray En Yvelines, France) was used as the model of progenitor cell differentiation to mature myelin-producing OLs. We chose the MO3.13 cell line, which differentiates after phorbol 12-myristate 13-acetate (PMA) stimulation, as the most adequate model of OPC maturation. Unlike other OL lines ( human oligodendrogliomaor KG-1C), MO3.13 cells during maturation exhibited the strongest similarity to primary human OLs in morphology as well as in gene and protein expression [27]. OLs were cultured in DMEM–high-glucose medium supplemented with 10% fetal bovine serum and 1% penicillin-streptomycin and maintained at 37 °C with 5% CO_2_ in humidified atmosphere. The cultures were conducted in 75-cm^2^ flasks (Nunc, Thermo Scientific). After 80% of confluence, cells were passaged using a 0.25% trypsin–EDTA solution in total three times for a week with a dilution factor of 1/8. To induce OPC maturation, PMA was added (0.1 mM) for 72 h, and cells were incubated at 37 °C with 5% CO_2_ in humidified atmosphere. All the reagents used for culture were purchased from Sigma-Aldrich. The purity and cell differentiation were estimated by oligodendrocyte marker O4, nestin, MOG, and glial fibrillary acidic protein (GFAP) expression by immunocytochemical (ICC) analysis, as well as morphological shape in differential contrast microscopy (DIC) examination.

### 2.3. hOPC Supplementation with Fish Oil Mixture

FOM derived from Centroscymnus crepitater, Etmopterus granulosus, Deania colceai, Centrophorus scalpratus, Sardinops sagax, Scomber scombrus, and Gadus morhua species (UPRP patent # P.416768) and linseed oil (LO) as a control were used to examine the effect of fish-derived oils and plant-derived oils on OPC differentiation and myelin synthesis. FOM contained: 17% of *cis*-5,8,11,14,17-EPA (C20:5n), 9% of *cis*-4,7,10,13,16,19-DHA (C22:6n3), 3% of squalene, and 13% of NA (C24:1n9). LO containing alpha-linolenic acid C18:3n3 (64.2%), palmitic acid C16:0 (2.2%), stearic acid C18:0 (3.7%), oleic acid C18:1n9c (16.5%), and linoleic acid C18:2n6c (13.4%) was used as control. The complete oil composition determined by gas chromatography-mass spectrometry is summarized in Appendix A.

At 24 h before the experiments, 2 × 10^6^ MO3.13 cells were seeded on six-well plates in DMEM–high-glucose medium supplemented with 5% FBS, 1% penicillin-streptomycin, at 37 °C with 5% CO_2_. PMA (0.1 mM) was used as a stimulator of OPC differentiation. In parallel experiments, 5% FOM or 5% LO were added to the culture for 24 to 72 h. After incubation, supernatants were collected and cells were washed in PBS for further analysis.

### 2.4. Lactate Dehydrogenase (LDH) Release Assay

LDH release was measured by the colorimetric method using a Cytotoxicity Detection Kit (Sigma-Aldrich, St. Louis, MO, USA) according to the manufacturer’s instruction. The concentration of released LDH from 100% of cell lysis (OLs lysed with 1%Triton X-100) was used as the positive control, and supernatants from OLs supplemented with 10% PBS were used as the negative one (spontaneous LDH release). The rate of lysed cells was calculated based on the normalization of each sample to the level of LDH released by the positive control subtracted from negative control samples. All the samples were analyzed in duplicate.

### 2.5. Immunocytochemical Analysis (ICC)

For quantitative immunofluorescence analysis, cells were transferred to gelatin-coated microscope slides by cytospin (300 x g, 10 min) and fixed with 4% PFA for 20 min at 21 °C. Fixed cells were washed with PBS and blocked with 10% rabbit blocking serum (Santa Cruz Biotechnology) supplemented with 3% Triton™ X-100 (Sigma-Aldrich) for 45 min at 21 °C. Cells were washed and double-stained for MOG/PLP MBP/nestin, O4/phalloidin, and GFAP/phalloidin. Anti-MOG, anti-MBP, anti-PLP (all from Santa Cruz Biotechnology), anti-O4 (R&D Systems), anti-Nestin (BioLegend), phalloidin (F-actin)/texas red (TR) (Invitrogen), anti-GFAP (Santa Cruz Biotechnology), and rat IgG2b (Invitrogen) were used as negative isotype controls. All the antibodies were suspended in PBS supplemented with 1.5% blocking rabbit serum, 0.3% Triton™ X-100, and 0.01% sodium azide, and incubated overnight at 4 °C. Cells were washed, and secondary fluorescent Abs were added for 1 h at RT: chicken pAbs to rabbit TR (Santa Cruz Biotechnology), goat pAbs to mouse fluorescein isothiocyanate (FITC) (Abcam), goat pAbs to mouse TR (Abcam). For DNA staining, 4′,6-diamidyno-2-fenyloindol (DAPI) (1.5 μg/mL UltraCruz™ Mounting Medium, Santa Cruz Biotechnology) was used. Images were acquired using a confocal microscope Nikon D-Eclipse C1 and analyzed with EZ-C1 v. 3.6 software. Fluorescence intensity was determined as the average fluorescence (average area), which was the sum of the fluorescence from all the segments divided by their number [28]. The average fluorescence was calculated using at least 100 single cells taken from four independent experiments. The level of baseline fluorescence was established individually for each experiment. Nonspecific fluorescence (signal noise) was electronically diminished to the level when a nonspecific signal was undetectable.

### 2.6. RNA Isolation and Total mRNA Concentration Analysis

Total RNA was extracted from human and OLs with a mirVanaTM miRNA Kit (Thermo Fisher Scientific). After isolation, the RNA level and the purity analysis were performed by an Agilent small RNA Kit (2100 Bioanalyzer, Agilent 2100 expert software). The data was shown as raw data of mRNA concentration (pg/mL).

### 2.7. Quantitative Real-Time PCR (qRT-PCR)

qRT-PCR was performed with TaqMan probes (ThermoFisher Scientific, Waltham, MA, USA) using a 7900 Real Time PCR System. First, 1 µg of total RNA was transcribed to cDNA using a SuperScript^®^ VILO™ cDNA Synthesis Kit (ThermoFisher Scientific, Waltham, Massachusetts, USA). cDNA was amplified in the presence of specific TaqMan probes—MBP, MOG, PLP, and β-actin—using TaqMan^®^ Fast Advanced Master Mix (ThermoFisher Scientific, Waltham, MA, USA). The PCR reactions were performed on the 96-well plates using TaqMan^®^ Fast Advanced Master Mix (ThermoFisher Scientific, Waltham, MA, USA). The reaction specificity was checked by melting curve analysis, and relative gene expression was determined by the ΔΔCT method.

### 2.8. Lipid Profiling for Free and Esterified Fatty Acids by Gas Chromatography-Mass Spectrometry (GC/MS)

For chromatographic lipid profile analysis, samples (brain or cultured cells) were lyophilized. Samples were frozen at −80 °C for 2 h, and then delivered to a FreeZone 18L lyophilizator (LABCONCO Kanasas City MO, USA) and lyophilized under vacuum conditions overnight (138Pa at −45 °C). To the lyophilized samples, 100 µL of methyl tert-butyl ether (Sigma-Aldrich) and 100 µL of a 0.25-M solution of trimethylsulfonium hydroxide in methanol (Sigma-Aldrich) were added. After that, the samples were incubated at 80 °C for 30 min, and gas chromatography/mass spectrometry was performed by GC/MS Thermo GC Ultra coupled with an ISQ™ mass spectrometer (Thermo Fisher Scientific, Waltham, USA). The components were directly injected into the port of a GC/MS. The GC columns Stabilwax (length 20 m, internal diameter 0.18 mm, film thickness 0.18 µm; from Restek Corp., Bellefonte, USA, cat. No. 40602) were used. Then, 1 µL of the sample was applied to the split/splitless (SSL) injector in splitless mode (injector temperature 240 °C). The GC oven temperature was initially held at 50 °C for 1 min, and then increased to 240 °C for 30 min at a rate of 4 °C/min. Helium as a carrier gas was used at a flow of 1 mL/min. Mass spectra were collected using a quadrupole mass spectrometer. The settings of mass spectrometry were as follows: ion source temperature 200 °C, ionization energy 70 eV, and scan range 33 to 650 amu (atomic mass unit). Obtained mass spectra were compared with NIST/EPA/NIH and Wiley Registry of Mass Spectral Data mass spectral libraries.

### 2.9. Cytokine, Chemokine, and Growth Factor Analysis by Human Cytokine Multiple Profiling Assays

Cytokine, chemokine, and growth factor concentrations in culture supernatants of MO3.13 cells and maturing hOPCs supplemented with 5% FOM or 5% LO or medium were measured using Bio-Plex Pro™ Human Cytokine Assays (Bio-Rad Laboratories, USA). Standards and samples were diluted (1:4) in sample diluent and transferred to the plate containing magnetic beads for 1 h at room temperature (RT). Next, the plate was washed (3 times), and a detection antibody was added for 30 min on a shaker (850 rpm) at RT. After that, the plate was washed (3 times) and streptavidin–phycoerythrin (PE) solution was added for 10 min. Subsequently, the plate was washed (3 times) and samples were re-suspended in 125 µL of assay buffer and analyzed within 15 min. All the samples were analyzed at the same time in duplicate. All the reagents and technology were provided by Bio-Rad Laboratories (Bio-Plex 200).

### 2.10. Statistics

Arithmetic means and standard deviations were calculated for all the parameters from at least four independent experiments. A statistical analysis of differences was performed using the one-way ANOVA test. Scheffe’s test was used for multiple comparisons as a post hoc test when statistical significances were identified in the ANOVA test. Statistical significance was set at *p* < 0.05.

## 3. Results

### 3.1. Dysregulated Lipid Composition in EAE Brain in Comparison to Healthy Mouse Brain

Progressive immune-mediated axonal loss and insufficient myelin production can result in different lipid composition in the EAE brain in comparison to a healthy control (HC) brain. We found that EAE brain lipid composition was characterized by a decreased rate of palmitic acid (C16:0), DHA (C22:6n3), and NA (C24:1n9). On the contrary, the percentage of arachidonic acid (AA) (C20:4n6) was significantly elevated about three times in the EAE brain. Moreover, an EAE brain contained relatively higher levels of oleic acid (C18:1n9c) and elaidic acid (C18:1n9t), contrary to a HC brain. In addition, several lipids were detected only in the EAE brain: palmitoleic acid (C16:1), pentadecanoic acid (C15:0), meristic acid (C14:0), linolelaidic acid (C18:2n6t), and *cis*-11-eicosenoic acid (C20:1n9). Other detectable brain lipids were not considerably different between EAE and HC brains (Table 1).

### 3.2. Maturating hOPCs Produce Nervonic Acid

The results of the whole brain lipid profiling suggested that upon the inflammatory conditions, the synthesis of NA is inhibited in favor of arachidonic acid (AA) (Figure 1A). The loss of DHA and NA during an acute phase of EAE should be compensated during the remyelinating phase. OLs are the principle cells that are responsible for forming neuron myelin sheaths. During the physiological reconstitution of CNS, OPCs, being abundant throughout the gray matter of CNS, migrate toward demyelinating plaque, where they can generate new OLs [29]. The structure of new myelin sheaths formed in remyelinating lesions was shown to be thinner than this during development [2], which can be the result of a dietary deficiency in DHA and NA in adults and persistent chronic inflammation in the plaques. To address this issue, we employed a model of maturating human OPCs (hOPCs) based on human MO3.13 cells stimulated with PMA that resulted in their differentiation into OLs. PMA-stimulated MO3.13 cells were incubated for 72 h, and lipid rates were analyzed at three time points (24 h, 48 h, and 72 h) (Figure 1B). First, we found that during the maturation of hOPCs, NA (C24:1n9) and its monounsaturated analog lignoceric acid (C24:0) synthesis was induced (Table 2). The biosynthesis of NA is initiated from oleic acid through fatty acid elongation catalyzed by malonyl-CoA, long-chain acyl-CoA, and 3-ketoacyl-CoA synthase [30]. We also noted a considerable decrease in the rate of palmitic acid (C16:0), stearic acid, and *cis*-13-docosenoic acid Me ester (C22:1n9) synthesis, which may serve as the substrates for NA synthesis or directly be incorporated into sphingosine (Figure 1C upper right panel). The correlation between newly formed lipids and a simultaneous decrease in other fatty acids reflected the cell metabolism analyzed at three time points (24 h, 42 h, and 72 h of incubation). Presumably, the constant NA synthesis initiated in OPCs is related to the cell readiness to the subsequent incorporation of NA into sphingomyelin [31].

### 3.3. hOPCs Incorporate and Metabolize Nervonic Acid

Next, we examined whether supplementation of the culture medium with FOM, in comparison to LO, would affect the lipid acid esters that are bound to sphingosine, creating sphingomyelin in the maturing hOPCs. PMA-stimulated MO3.13 cells (hOPCs) were incubated alone or in the presence/absence of FOM at a final concentration of 5% or LO used as a control for 72 h. In the group of detected lipids, we noted three of the most important lipid acid esters being substrates for the synthesis of sphingomyelin: nervonic, palmitic, and stearic acid esters (Table 2). We found at 24 h of incubation that the percentage of NA ester in hOPCs was increased six to seven times in the presence of 5% FOM, in comparison to hOPCs incubated alone (Figure 1C and Table 2). Palmitic and stearic acid esters are constitutively presented in hOPCs and can be directly used for sphingomyelin synthesis as well as for NA synthesis with *cis*-13-dicosenoic acid as an intermediate product (Figure 1A,C). During FOM supplementation, NA was directly incorporated by hOPCs, while the *cis*-13-dicosenoic acid-dependent pathway was inhibited. At 48 h, the rate of NA ester dropped dramatically, suggesting its metabolism during hOPC maturation. At 72 h, NA was undetectable in hOPCs in the presence of FOM, while there were no changes observed during the incubation without FOM. The loss of NA might result from its binding via an amide bound to a sphingosine base [31,32] during sphingolipid synthesis by finally differentiated OLs [33].

### 3.4. FOM Affects Maturing hOPCs Resulting in Increased Production of Myelin Proteins

Fish oils may improve myelin synthesis by providing essential substrates as well as by facilitating a specific anti-inflammatory environment. We showed a simultaneous increase in MOG, PLP, and MBP production by hOPCs in the presence of 5% FOM, at the protein (Figure 2) and mRNA level (Figure 3) in comparison to LO. A time-lapse analysis of myelin protein mRNAs demonstrated their highest values at 48 h of incubation for MOG and PLP with a subsequent decrease, whereas MBP mRNA was only moderately and constantly increased after 72 h of incubation (Figure 3A). Moreover, we found that FOM supplementation resulted in an increased total mRNA level in hOPCs, suggesting its boosting effect on the cell metabolism (Figure 3B).

### 3.5. FOM Affects hOPC Production of Cytokines, Chemokines and Growth Factors

Maturating hOPCs colonizing demyelinating lesions produce several mediators, including growth factors, chemokines, and cytokines, which in physiological conditions (without autoreactive lymphocytes and chronic inflammation) create the environment responsible for CNS regeneration [5]. Therefore, using a cytokine multiple profiling assay, we have analyzed supernatant concentrations of 27 cytokines, chemokines, and growth factors. We revealed that hOPCs cultured with FOM in comparison to hOPCs cultured in medium produced statistically significant increased amounts of fibroblast growth factor 2 (FGF-2) and vascular endothelial growth factor (VEGF), as well as decreased amounts of interleukin (IL)-5, IL-6, IL-7, IL-8, IL-15, IL-17, eotaxin, granulocyte colony-stimulating factor (G-CSF), interferon (IFN)-γ, monocyte chemoattractant protein 1 (MCP-1)*,* and RANTES (Regulated on Activation, Normal T-cell Expressed and Secreted) (Table 3). Conversely, LO supplementation had no effect on IL-5, IL-6, IL-7, IL-8, IL-15, IL-17, eotaxin, FGF-2, MCP-1, RANTES, and VEGF production by hOPC (Table 3). Taken together, FOM supplementation in contrast to LO supplementation tends to inhibit proinflammatory cytokine/chemokine production by hOPCs and enhances the growth factors involved in CNS regeneration.

### 3.6. Fish Oil Mixture Promotes hOPC Survival and Prolongs OL Lifespan

The previous set of experiments demonstrated that myelin protein mRNA was significantly dropped in the presence of LO. Thus, we checked the effects of both types of oils on hOPC viability and apoptosis. We found that fish-derived oil was not cytopathic for hOPCs, which was confirmed by LDH release and annexin V/iodine propide (ANX-V/PI) labeling assays (Figure 4A,B). A normalization of sample concentrations to LDH levels in positive and negative controls revealed that approximately 5–7% of OLs underwent lysis in the presence of FOM at three time points (24 h, 42 h, and 72 h), whereas LO caused cell lysis of 16%, 24%, and 26%, respectively (Figure 4B). Moreover, we observed the decreased rate of hOPC apoptosis in the presence of FOM, while LO markedly increased both apoptosis and LDH levels (Figure 4A,B). DIC microscopy analysis presented that FOM, contrary to LO, had no effect on hOPC morphology (Figure 4C).

## 4. Discussion

Neurogenesis induced in an attempt of brain self-repair might be a potential therapeutic target aiming at the differentiation and survival of neural cells. It is assumed that limited remyelination during the remission of MS may be associated with an incapability of OPCs to repopulate the area of demyelination and/or result from the presence of inhibitory factors and/or lack of stimuli required to generate remyelinating OLs in the area of demyelination [35].

In this study, we demonstrated that remyelination can be affected by the presence of fish oils rich in EPA, DHA, and NA in the microenvironment. We found that maturing hOPCs cultured in the presence of FOM incorporated and metabolized NA, and transformed into OLs characterized by normal morphology and prolonged lifespan. These OLs synthetized increased amounts of the myelin proteins MOG, PLP, and MBP. Conversely, LO that contained primary n-3 alpha-linolenic acid ester (C18:3n3) used as the control negatively affected hOPC culture, increasing cell apoptosis and lysis. Similar results were demonstrated by in vitro study with the rat oligodendrocyte progenitors supplemented with DHA [36], and an in vivo study using neonatal rats with the intracerebroventricular administration of DHA or EPA what promoted myelinogenesis and enhanced PLP and MBP expression [37]. The molecular mechanisms underlying these effects might be based on the modulation of cAMP intracellular levels and the activity of the transcription factor cAMP response element binding (CREB) [37], but also on the activation of peroxisome proliferator-activated receptor-γ and the phosphorylation of extracellular signal-regulated kinase (ERK) 1-2 [36]. The activation of ERK1/2 has been found to be a crucial step in OPC transition to OL stage [38].

The whole brain profiling of lipid composition in EAE, an animal model of MS, in comparison to healthy mice, revealed two major changes that might be critical for the disease progression. The first one was related to the dysregulation and decrease of NA synthesis, and the second one was related to an increased synthesis of AA (C20:4n3). As both NA and AA synthesis pathways are based on the same substrates, the switch in the lipid production might be mediated by an inflammatory environment in the brain. In the EAE brain, we detected two specific lipids, linolelaidic acid (C18:2n6t) and *cis*-11-eicosenoic acid (C20:1n9), both serving as substrates for AA synthesis [39]. During maturation to myelin-producing OLs, hOPCs induce NA synthesis. In the in vitro model of hOPC maturation using the human MO3.13 cell line stimulated by PMA, the main source of NA synthesis came from the palmitic (C16:0)/stearic (C18:0)/oleic (C18:1n9c) acid ester pathway. Another product involved in NA synthesis is a linoleic acid ester (C18:2n6c), which is catalyzed by Δ6 desaturase for transformation into oleic acid (C18:1n9c). There is also the direct possibility of DHA (C22:1n9) transformation into NA (C24:1n9) (Figure 1A red arrows). Considering the results from the chromatographic analysis of these two experimental settings and the synthesis of n-9 fatty acids in humans being low compared to other species [4], an intake of exogenous NA could support the process of remyelination by improving sphingolipid synthesis by OLs [21].

The supplementation of maturing hOPCs with FOM resulted in the enhancement of MOG and PLP, and to a lesser degree in MBP synthesis. The dynamics of NA fluctuation against other lipid profile in OPCs suggest the influence of NA on the metabolic cell rate. Our study demonstrated that FOM supplementation of culture medium during hOPC maturation significantly affected the transcription process, which was expressed as a several-fold increase in the total level of mRNA compared to non-supplemented culture. Furthermore, increased mRNA and protein synthesis of the three most essential myelin proteins—MOG, PLP and MBP—were observed. The microscopy analysis of OL morphology as well as the prolonged lifespan, which was assessed as apoptosis and the cell lysis rate decreased, may indicate that supplementation of OPCs with FOM, contrary to LO, improved OPC survival and maturation in the culture. The molecular mechanisms of these effects are obscured; however, the literature data suggests two possibilities: interaction of NA with DNA polymerase β enzymes (DNA Pol II) [40], and activation of the β-oxidation pathway [41,42]. The first mechanism is mediated by DNA Pol lI, which is involved in the regulation of DNA maintenance and cell replication. The second one is associated with oxidative metabolism regulated by two enzymes, AMP-activated protein kinase (AMPK) [43] and sirtuins (SIRT), as well as peroxisome proliferator-activated nuclear receptors (PPARs) [41,42]. The ligation of PPAR results in the upregulation of gene transcription involved in the β-oxidation of fatty acids, and together with AMPK/sitrtuins, forms networks. This dyad is regulated by physical conditions as well as a diet rich in the long-chain polyunsaturated fat acids that are inherent in fish oil [44].

Our data are also in the line with PUFA dietary intervention trials in humans. Fish-derived oils possess neuroprotective effects at many levels. In vitro studies demonstrated that DHA and EPA promoted the differentiation of neural stem cells into neurons [20,45]. The regulatory effects of both lipids are also associated with neural stem cell differentiation-mediated divergent effects via transcription factors and different signaling pathways [13]. DHA and EPA were additionally demonstrated to exert anti-inflammatory properties, in in vitro experiments with peripheral mononuclear cells [46].

Our previous data suggest that fish oil formula was also composed to have an immunomodulatory effect that is associated with biological properties of PUFAs and squalene [47,48]. In this study, we demonstrated that during the differentiation of hOPCs to myelin protein-producing OLs, these cells synthetize proinflammatory IL-5, IL-6, IL-7, IL-8, IL-9, IL-15, IL-17, eotaxin, IFN-γ, IP-10, MCP-1, MIP-1α, and RANTES. Markedly increased IL-6 and IL-8 synthesis might result in the recruitment and activation of proinflammatory cells in the brain [49]. When cell cultures were supplemented with FOM, opposite to LO, a significant decrease in proinflammatory cytokine/chemokine production was noted. Simultaneously, we observed an up-regulation of FGF-2 and VEGF secretion in the culture of maturing hOPCs supplemented by FOM. During physiological CNS reconstitution, FGF-2 is synthetized by perivascular astrocytes and residual microglia/macrophages, as well as by OPCs themselves [6,50]. It acts as a chemoattractant for OPCs, expressing FGF receptors (1–4), and orchestrates MBP synthesis by mature OLs [6,51]. The role of VEGF seems to be pleiotropic. Although VEGF acts as a proinflammatory factor in the early phase of CNS inflammation, its reduced reactivity in the late phase can result in the dysregulation of neural progenitor proliferation, migration, differentiation, and OPC survival and migration to demyelinated lesions [52].

One of the limitations in our study is the use of an MS animal model to estimate fluctuations in the lipid composition, and subsequent speculation about the inhibition of NA synthesis in MS patients. Although the EAE model accurately reflects the main pathological processes occurring during MS, human brain tissue has a significantly greater lipidome divergence compared to mice [53]. Another limitation is the use of fish-derived oil mixture instead of isolated NA. Natural FOM was chosen to mimic dietary conditions where a range of essential oils is consumed. The changes in the lipid metabolism may depend on the environmental conditions and the availability of substrates. As multiple synergetic effects of fish-derived EPA and DHA were already documented [9,11,15,46,47,48], we speculate that the NA separated function would not be so beneficial and complete as an effect of a specially prepared oil formula. Additionally, we failed to demonstrate any improvement of hOPCs functions in the cultures supplemented with LO, which were characterized by a lack of NA but the presence of n-3 PUFAs.

In conclusion, our findings indicate that during EAE, lipid metabolism in the brain is redistributed from the synthesis of NA to AA. The supplementation of hOPC culture medium with fish oil rich in EPA, DHA, and NA improves the ability of mature OLs to synthesize myelin proteins as well as sphingomyelin. In addition, FOM inhibits the secretion of several proinflammatory factors by hOPCs that can be potentially involved in the immune cell recruitment during remyelination, and promotes growth factor synthesis, which is important for the regeneration of CNS. Future studies on the animal model should demonstrate the ability of FOM compounds to cross the BBB and determine its usefulness in the context of brain remyelination.

## 5. Patents

Fish oils mixture (FOM) is restricted by patent no. UPRP P.416768.

## Figures and Tables

**Figure 1 cells-08-00786-f001:**
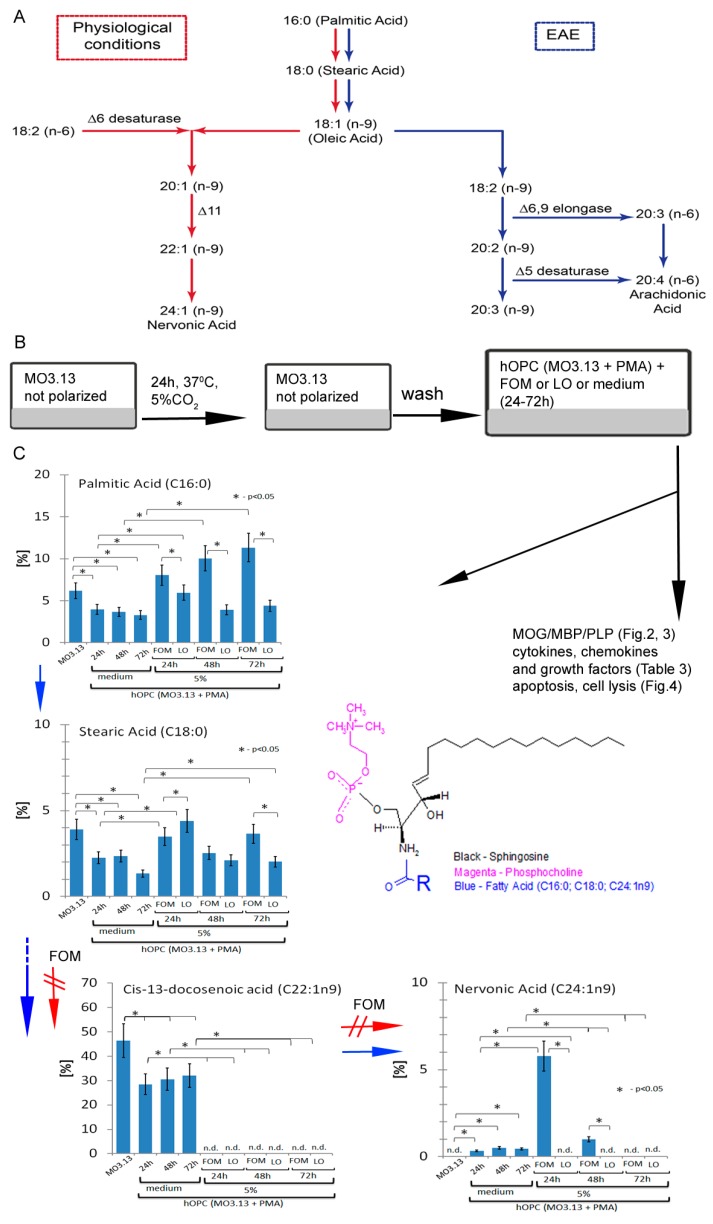
Human oligodendrocyte precursor cells (hOPCs) incorporate and metabolize nervonic acid. (**A**) GC/MS lipid profiling of the whole brain tissue revealed that nervonic acid (NA) ester (red arrows) is not synthetized during EAE in contrast to arachidonic acid ester (blue arrows) known for its proinflammatory properties. (**B**) Human phorbol 12-myristate 13-acetate (PMA)-stimulated MO3.13 cells, as the cellular model of oligodendrocyte progenitor cells (OPC) polarization to mature myelin-producing oligodendrocytes, were used to determine an effect of fish oil mixture (FOM) supplementation on OPC function. LO was used as the negative control. PMA-stimulated MO3.13 cells were incubated with 5% FOM or 5% LO for 72 h. (C middle right panel) Sphingomyelin consists of a phosphocholine head group, sphingosine, and fatty acid esters which are bound via an amide bound to a sphingosine base [31,34]. Fatty acid esters were detected by the GC/MS method. (**C**) NA, opposite to palmitic and stearic acid is synthetized *de novo* by maturating OPCs. Palmitic and stearic acid esters are constitutively presented in hOPCs, and can be directly used for sphingomyelin synthesis or can also be used as substrates for NA synthesis with *cis*-13-dicosenoic acid as an intermediate product (blue arrows). During FOM supplementation, NA is directly incorporated by hOPCs, while the *cis*-13-dicosenoic acid-dependent pathway is inhibited (red arrow). The rates of acid esters were measured at three time points (24 h, 48 h, and 72 h) in lyophilisates of hOPCs incubated with FOM or LO or in medium. Data are presented as means ± SD from four independent experiments.

**Figure 2 cells-08-00786-f002:**
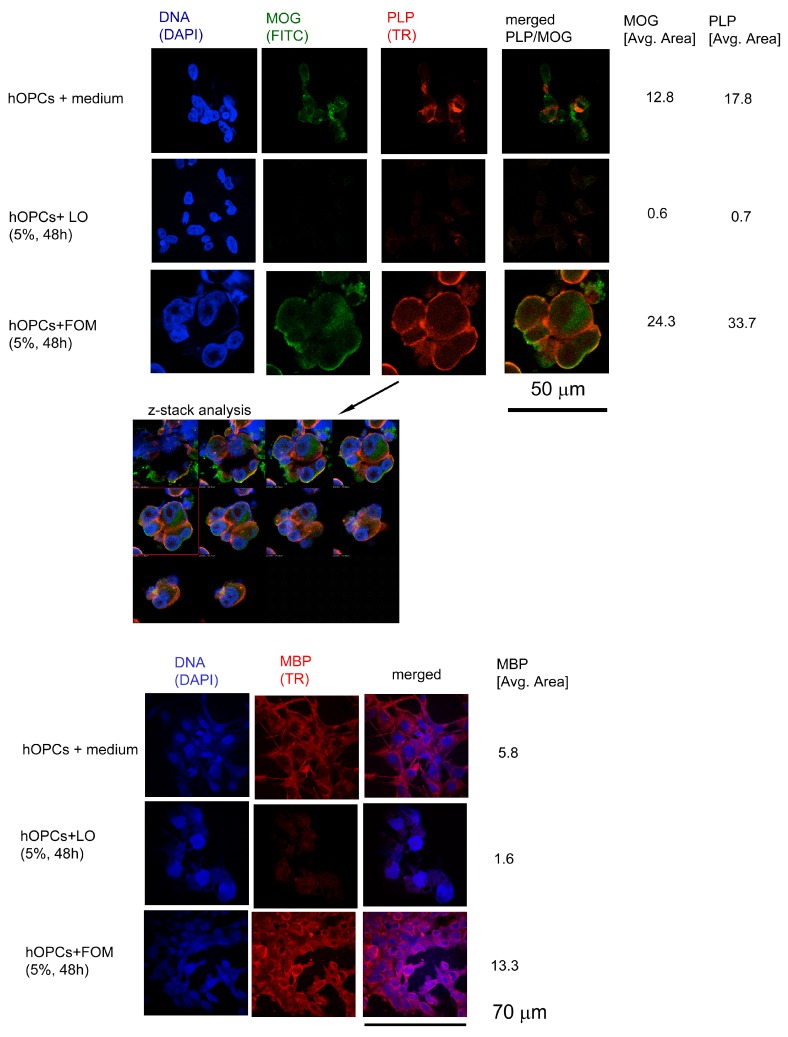
Supplementation with FOM opposite to LO during hOPC maturation enhances myelin oligodendrocyte glycoprotein (MOG), proteolipid protein (PLP), and myelin basic protein (MBP) synthesis by mature oligodendrocytes (OLs). Immunocytochemical (ICC) characterization of hOPCs supplemented with FOM during their differentiation into mature MBP/PLP/MOG-producing OLs. Confocal z-stack analysis confirmed that hOPCs supplemented with FOM were characterized by MOG/PLP accumulation in the intracellular space during their maturation. The data from the quantitative fluorescence signal of protein analysis are presented as average area (Avg Area) from four independent experiments.

**Figure 3 cells-08-00786-f003:**
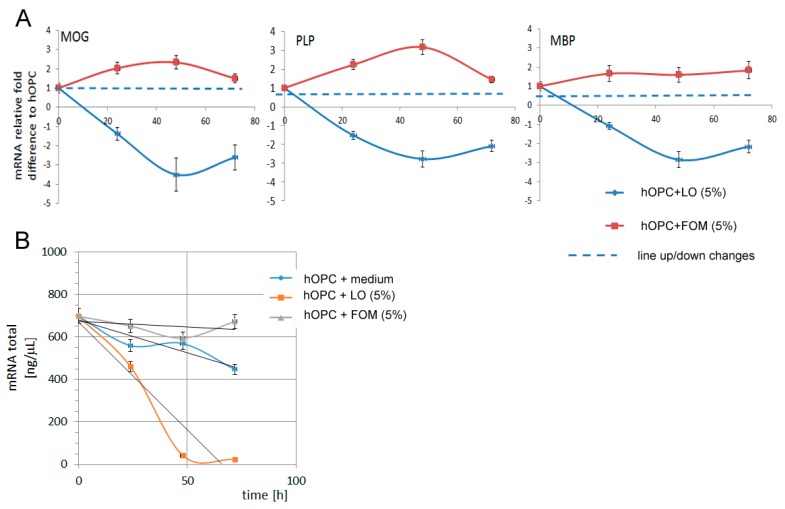
(**A**) Quantitative analysis of myelin protein mRNA expression confirmed that hOPCs supplemented with FOM contrary to LO were characterized by increased MOG, PLP, and MBP synthesis during maturation into OLs. (**B**) Increased global mRNA levels in hOPCs supplemented with FOM. Data are presented as means ± SD from four independent experiments.

**Figure 4 cells-08-00786-f004:**
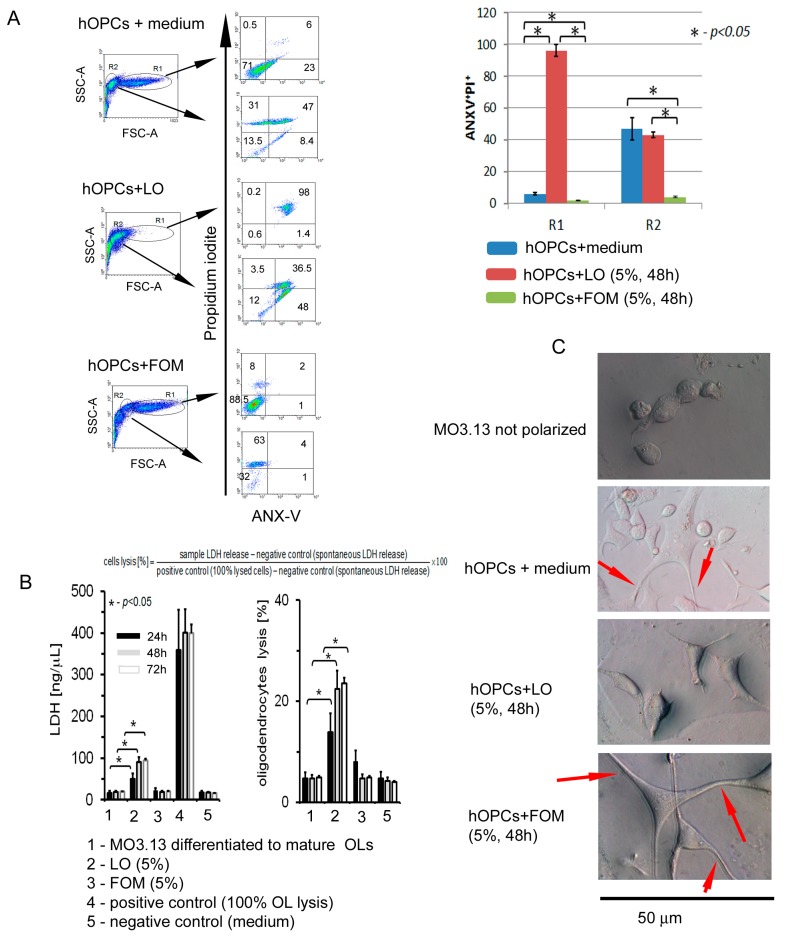
Fish oil mixture prolongates OL lifespan. (**A**) FOM supplementation inhibited spontaneous apoptosis of OLs. Two regions, R1 and R2, were gated based on the morphology (FSC vs. SSC) of MO3.13 cells and analyzed by flow cytometry. The rates of cells undergoing a late phase of apoptosis (double-positivity for propidium iodine and Anexin V) are presented as means ± SD from four independent experiments. (**B**) FOM supplementation had no effect on OL lysis, as LDH concentrations in supernatants were not statistically significant compared to hOPCs cultured alone (left panel). Data (means ± SD from four independent experiments) are presented as LDH concentrations and changes were normalized to positive and negative controls (right panel) according to the formula presented in the upper panel (percentage of lysed oligodendrocytes). (**C**) Differential contrast microscopy (DIC) microscopy analysis presented that FOM, contrary to LO, had no effect on hOPC morphology. Red arrows indicate large lamellipodia and filipodia forming a branched network, confirming the typical structure of OLs in the presence of FOM.

**Table 1 cells-08-00786-t001:** The lipid composition of experimental autoimmune encephalomyelitis EAE and healthy control (HC) brains. Red color indicates the lipids that are reduced in the course of disease. A blue color indicates the lipids that are relatively upregulated during EAE, while a green color indicates the new appearing lipids involved in arachidonic acid (AA) metabolism. Data are presented as the percentage of lipids from three independent experiments (one mouse per one experiment).

	Lipid Acid	HC [%]	EAE [%]
1	Myristic Acid (C14:0)	n.d.	0.13
2	Palmitic Acid (C16:0)	39.48	35.14 *
3	Palmitoleic Acid (C16:1)	n.d.	0.65
4	Pentadecanoic Acid (C15:0)	n.d.	0.58
5	Stearic Acid (C18:0)	18.85	18.06
6	Oleic Acid (C18:1n9c)	18.19	20.30
7	Elaidic Acid (C18:1n9t)	5.92	8.16 *
8	Linolelaidic Acid (C18:2n6t)	n.d.	0.83
9	*cis* -11-Eicosenoic Acid (C20:1n9)	n.d.	0.66
10	Arachidonic Acid (C20:4n6)	1.74	5.56 *
11	*cis* -4,7,10,13,16,19-Docosahexaenoic Acid (C22:6n3)	14.80	9.76 *
12	Nervonic Acid (C24:1n9)	1.91	n.d. *

n.d.—non detectable (<0.1%); * *p* < 0.05.

**Table 2 cells-08-00786-t002:** The effect of FOM and linseed oil (LO) on the lipid composition of maturing hOPC. The red color indicates the main lipids that are involved in NA synthesis or directly bind to sphingomyelin during hOPC maturation. Data are presented as the percentage of lipids from four independent experiments.

	Lipid Acid	MO3.13 Cells not Polarized			hOPCs (PMA)			hOPCs (PMA) + FOM			hOPCs (PMA) + LO		
		24 h	48 h	72 h	24 h	48 h	72 h	24 h	48 h	72 h	24 h	48 h	72 h
**1**	Myristic acid (C14:0)	n.d.	n.d.	n.d.	n.d.	n.d.	n.d.	1.15#	1.20#	1.85#	n.d.	n.d.	0.12
**2**	Pentadecanoic acid (C15:0)	n.d.	n.d.	n.d.	n.d.	n.d.	n.d.	n.d.	0.13	n.d.	n.d.	n.d.	n.d.
**3**	Palmitic acid (C16:0)	9.58	9.28	8.78	3.98 †	3.66 †	3.31 †	8.04 #	10.04 #	11.32 #	5.96	3.91	4.42
**4**	Palmitoleic acid (C16:1)	n.d.	n.d.	n.d.	n.d.	n.d.	n.d.	2.48#	3.55#	3.2#	n.d.	n.d.	n.d.
**5**	Heptadeconoic acid (C17:0)	n.d.	n.d.	n.d.	n.d.	n.d.	n.d.	0.41	0.77	n.d.	n.d.	n.d.	n.d.
**6**	Heptadeconoic acid, 16-methyl	n.d.	n.d.	n.d.	n.d.	n.d.	n.d.	n.d.	0.10	n.d.	n.d.	n.d.	n.d.
**7**	*Cis*-10-heptadecenoic acid (C17:1)	n.d.	n.d.	n.d.	n.d.	n.d.	n.d.	n.d.	1.67#	n.d.	n.d.	n.d.	n.d.
**8**	Stearic acid (C18:0)	4.89	5.33	4.09	1.05 †	1.01 †	1.34 †	2.82 #	2.53 #	3.65 #	4.44 #	2.10	2.06
**9**	Elaidic acid (C18:1n9t)	0.7	3.89	2.57	n.d.	n.d.	n.d.	22.49 #	23.12 #	24.33 #	17.04 #	16.81 #	16.34 #
**10**	Oleic Aaid (C18:1n9c)	17.98	20.21	17.65	17.87	18.90	18.86	n.d. #	n.d. #	n.d. #	n.d. #	n.d. #	2.24 #
**11**	Linoleic acid (C18:2n6c)	19.46	12.61	23.52	33.28 †	30.58 †	28.28 †	8.46 #	1.42 #	3.47 #	14.49 #	28.14	25.93
**12**	Linolelaidic acid (C18:2n6t)	n.d.	n.d.	2.12	n.d.	3.18	3.78	0.58 #	0.16 #	n.d. #	n.d.	2.41	0.29 #
**13**	Gamma-Linolenic acid (C18:3n6)	n.d.	n.d.	n.d.	3.4 †	2.34 †	2.38 †	n.d. #	n.d. #	n.d. #	n.d. #	n.d. #	0.67 #
**14**	Hexadecanoic acid	n.d.	n.d.	n.d.	n.d.	n.d.	n.d.	n.d.	0.03	n.d.	n.d.	n.d.	n.d.
**15**	Alpha-Linolenic acid (C18:3n3)	n.d.	n.d.	0.66	6.82 †	5.48 †	4.58 †	2.72 #	0.82 #	n.d. #	54.71 #	6.66	13.05 #
**16**	Stearidonic acid (C18:4n3)	n.d.	n.d.	n.d.	0.13	0.18	0.18	0.821	1.89	3.54	n.d.	n.d.	n.d.
**17**	*Cis* -11-Eicosenoic acid (C20:1n9)	1.05	1.35	1.19	3.74 †	2.85 †	2.82 †	5.11	5.46 #	5.93 #	n.d #	2.32	1.94
**18**	*Cis*-11,14,17-Eicosatrienoic acid (C20:3n3)	n.d.	n.d.	n.d.	n.d.	n.d.	n.d.	n.d.	n.d.	n.d.	n.d.	n.d.	0.27
**19**	*Cis*-5,8,11,14,17-Eicosapentaenoic acid (C20:5n3)	n.d.	n.d.	n.d.	n.d.	n.d.	n.d.	9.82 #	10.53 #	12.57 #	n.d.	n.d.	n.d.
**20**	*Cis*-8,11,14-eicosatrienoic acid (C20:3n6)	n.d.	n.d.	n.d.	n.d.	n.d.	n.d.	n.d.	0.05	n.d.	n.d.	n.d.	n.d.
**21**	Arachidic acid (C20:0)	n.d.	n.d.	n.d.	n.d.	n.d.	n.d.	n.d.	0.23	0.25	n.d.	0.64	0.51
**22**	Arachidonic acid (C20:4n6)	n.d.	n.d.	n.d.	n.d.	n.d.	n.d.	0.45	0.94	1.21 #	n.d.	n.d.	n.d.
**23**	Behenic acid (C22:0)	n.d.	n.d.	n.d.	n.d.	n.d.	n.d.	n.d.	0.09	n.d.	n.d.	1.74 #	0.97 #
**24**	Cetoleic acid (C22:1n11)	n.d.	n.d.	n.d.	n.d.	n.d.	n.d.	16.02 #	14.26 #	13.04 #	3.34 #	35.22 #	30.4 #
**25**	Docosapentaenoic acid	n.d.	n.d.	n.d.	n.d.	n.d.	n.d.	n.d.	1.65	n.d.	n.d.	n.d.	n.d.
**26**	*Cis* -13-docosenoic acid (C22:1n9)	46.38	47.27	38.74	28.51 †	30.58 †	32.06 †	n.d. #	n.d. #	n.d. #	n.d. #	n.d. #	n.d. #
**27**	*Cis*-4,7,10,13,16,19-Docosahexaenoic acid (C22:6n3)	n.d.	n.d.	n.d.	n.d.	n.d.	n.d.	5.34 #	9.18 #	7.52 #	n.d.	n.d.	n.d.
**28**	Lignoceric acid (C24:0)	n.d.	n.d.	n.d.	0.39	0.17	0.38	n.d.	n.d.	n.d.	n.d.	n.d.	0.26
**29**	Nervonic acid (C24:1n9)	n.d.	n.d.	n.d.	1.32	1.31	1.44	6.78 #	1.63 #	n.d. #	n.d. #	n.d. #	n.d. #
**30**	Squalene	n.d.	n.d.	n.d.	n.d.	n.d.	n.d.	5.78 #	9.05 #	8.08 #	n.d.	n.d.	n.d.

†—statistically significant differences between the medium of MO3.13 cells and the medium of hOPCs; #—statistically significant differences between the medium of hOPCs and FOM or LO hOPCs; n.d.—non detectable (<0.03%).

**Table 3 cells-08-00786-t003:** The profile of cytokines, chemokines, and growth factors released by MO.3.13 cells and hOPCs supplemented with FOM or LO. Data are presented in pg/mL as means ± SD from four independent experiments.

	Medium (48 h)		FOM (5%, 48 h)		LO (5%, 48 h)	
	MO3.13	hOPC	MO3.13	hOPC	MO3.13	hOPC
IL-1β	0.1 ± 0.04	0.3 ± 0.29	0.1 ± 0.06	0.2 ± 0.16	0.1 ± 0.05	0.2 ± 0.13
IL-1ra	30.1 ± 13.66	40.8 ± 17.97 †	33.5 ± 17.59	40.0 ± 19.94	28.3 ± 12.00	37.2 ± 18.31
IL-2	n.d.	n.d.	n.d.	n.d.	n.d.	n.d.
IL-4	n.d.	n.d.	n.d.	n.d.	n.d.	n.d.
IL-5	6.6 ± 2.90	17.3 ± 9.56 †	6.5 ± 3.55	9.9 ± 3.92 #	11.0 ± 5.44	16.0 ± 11.80
IL-6	5.4 ± 0.99	22.9 ± 3.93 †	4.0 ± 0.79	7.1 ± 1.02 #	4.9 ± 1.11	19.6 ± 4.02
IL-7	13.1 ± 9.81	24.5 ± 11.32 †	10.6 ± 6.61	11.0± 5.58 #	12.0 ± 10.82	21.7 ± 15.09
IL-8	358.4 ± 52.62	1102.6 ± 167.59 †	419.6 ± 59.55	651.2 ± 91.05 #	535.6 ± 84.53 *	1287.9 ± 192.67
IL-9	6.1 ± 2.98	13.3 ± 9.33 †	6.1 ± 2.11	9.8± 5.57	8.1 ± 3.97	12.4 ± 9.11
IL-10	6.6 ± 2.04	8.1 ± 3.98	6.4 ± 2.62	6.2 ± 2.77	6.3 ± 2.96	5.7 ± 3.99
IL-12	n.d.	n.d.	n.d.	n.d.	n.d.	n.d.
IL-13	1.1 ± 0.88	1.7 ± 0.55	1.3 ± 0.42	1.6 ± 0.71	1.2 ± 0.51	1.4 ± 0.56
IL-15	36.4 ± 16.82	58.5 ± 17.07 †	20.7 ± 12.77 *	9.7 ± 7.88 #	42.7 ± 21.29	53.2 ± 28.67
IL-17A	4.8 ± 2.95	7.1 ± 3.56 †	4.5 ± 2.11	4.1 ± 1.46 #	4.1 ± 3.34	7.3 ± 3.86
Eotaxin	54.4 ± 21.33	90.9 ± 32.78 †	36.6 ± 16.73 *	49.0 ± 20.55 #	31.4 ± 20.76 *	72.3 ± 22.51
FGF-2	n.d.	18.6 ± 4.33 †	16.9 ± 3.21 *	40.9± 6.81 #	14.2 ± 2.77 *	18.5 ± 2.80
G-CSF	95.9 ± 46.15	132.1 ± 52.82 †	94.0 ± 26.82	62.0 ± 12.90 #	90.1 ± 26.80	78.3 ± 21.91 #
GM-CSF	n.d.	0.6 ± 0.46	n.d.	0.6 ± 0.29	n.d.	0.8 ± 0.60
IFN-γ	50.2 ± 12.96	110.1 ± 25.44 †	44.9 ± 22.39	54.4 ± 24.86 #	39.5 ± 15.91	72.7 ± 22.55 #
IP-10	33.4 ± 18.80	53.4 ± 21.96 †	44.9 ± 24.62	43.7 ± 22,95	40.2 ± 18.49	46.8 ± 21.88
MCP-1	1175.4 ± 201.60	2105.6 ± 399.63 †	857.8 ± 142.18 *	968.2 ± 199.33 #	976.9 ± 247.78	1567.0 ± 451.83
MIP-1α	n.d.	n.d.	n.d.	n.d.	n.d.	n.d.
PDGF-bb	56.7 ± 17.97	82.7 ± 23.55 †	53.5 ± 10.04	79.4 ± 12.18	53.6 ± 19.76	79.6 ± 21.68
MIP-1β	2.8 ± 1.13	6.9 ± 2.99 †	4.3 ± 2.98	5.2 ± 3.08	3.6 ± 1.83	6.0 ± 2.75
RANTES	16.4 ± 9.33	28.3± 9.74 †	15.4 ± 10.59	14.2 ± 9.44 #	16.1 ± 8.07	23.7 ± 12.94
TNF	n.d.	n.d.	n.d.	n.d.	n.d.	n.d.
VEGF	99.2 ± 21.69	139.6 ± 43.80 †	119.6 ± 28.75	227.9 ± 69.43 #	110.3 ± 41.54	140.0 ± 59.43

†—statistically significant differences between medium of MO3.13 cells and medium of hOPCs; *—statistically significant differences between medium of MO3.13 cells and FOM or LO MO3.13 cells; #—statistically significant differences between medium of hOPCs and FOM or LO hOPCs; n.d. —non-detectable.

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
