# Peer review of "Naturally Occurring Nervonic Acid Ester Improves Myelin Synthesis by Human Oligodendrocytes"

_cells, 2019, doi:10.3390/cells8080786_

Round 1

Reviewer 1 Report

The authors studied the lipid composition in the brain tissues of normal and EAE mice and discovered reduced rate of DHA and NA while the percentage of AA was increased. Furthermore, a cell line of human oligodendrocyte precursor cells showed to produce NA during PMA induced maturation. And FOM improves myelin synthesis of these cells in vitro.  The authors should  include statistical information for all figures and tables and revise text to reflect the significance of these findings. 

Specific comments:

1.      Figure legends should be placed near the relevant figures. Figure 3 legend was placed underneath figure 2, and it was difficult to know which is figure 2 and which is figure 3 or figure 4.

2.      Tables 1 and 2, present data in Mean±SEM, also include statistical analysis to indicate whether the changes are statistically significant.

3.      Although your control is the cell line without polarisation with PMA, the culture condition should be the same.  In Figure 1B, after washing, the 3rd box should also include control cell line cultured without FOM or LO for 24-72h?

4.      Figure 1C, add statistical information.

5.      LO seems to have negative effect on hOPC survival and growth, is it the best control for your study?

6.       Lines 262-276, need to include your data to match the description in the text. For example, line 268, “… was increased 6 to 7 times…” where is the data? Did you mean Figure 1C? Does the data from Table 2 also support this? Need to present clearly.

7.      Cytokine data should include proper statistical information in the text and figure.

8.      Abbreviations are not used properly, here are some examples:

BBB, line 64 should be “… blood brain barrier (BBB)…”.

NA, lines 74, 118 and several other places.

PFA in line 94 should include full name.

PMA, LO, FOM should be used instead of the full name in several places, e.g. lines 99, 115 and 129.

Did you introduce what the full names are for AA and HC (line 230)?

Check all abbreviations throughout the paper.

Add matching abbreviations in Tables and Figures.

9.      Line 122, “lost” should be replaced with “reduced”?

10.  All “in vitro” should be in italic style, e.g. line 24.

11.  Is C57B/6 correct? Should it be C57BL/6?

12.  Line 70, check “… a mark neurodegeneration…”

13.  Line 107, check “..0,7mM..”, should it be  “..0.7mM..”?

14.  Line 411, check “theirselves”.

15.  Check the sentences in Lines 415-422, description is not clear.

Author Response

1.      The inappropriate placement of the figures and figure legends was performed during editing the text by the editorial office. Initially, each figure with the relevant legend was submitted on the separate sheet.

2.      The statistics have been added to the tables.

3.      The control cells were MO3.13 cells polarized with PMA (hOPCs) incubated in the culture medium alone. The appropriate changes have been made in the Figure 1 to clarify this issue.

4.      The statistics have been added to the Figure 1C.

5.      The idea to use linseed oil initiated from the results in the EAE mice demonstrating reduced concentration of NA in the brain. Therefore, we decided to supplement OPCs with the fish oil rich in NA and PUFAs n-3 in comparison to the plant-derived oil rich in PUFAs n-3 as both are considered to be a part of healthy diet (see Supplementary Table 1). We were surprised to observe negative effect of LO on hOPCs.

6.      The data is supported by the Figure 1C (mistakenly sited as 1B).

7.      We have included new results of 27 cytokines, chemokines and growth factors from multiplex assay (we have added Table 3 instead of Figure 4). The appropriated statistics have been done with the description in the main text.

8-14. We have checked and corrected the abbreviations and other language mistakes.

15. We have corrected the sentences in lines 415-422 to be more understandable:

‘In conclusion, our findings indicate that during EAE, lipid metabolism in the brain is redistributed from the synthesis of NA to AA. Supplementation of hOPC culture medium with fish oil rich in EPA, DHA and NA improves the ability of mature OLs to synthesize myelin proteins as well as sphingomyelin. In addition, FOM inhibits secretion of several pro-inflammatory factors by hOPCs that can be potentially involved in the immune cell recruitment during remyelination, and promotes growth factor synthesis important for the regeneration of CNS. The future studies on the animal model should demonstrate the ability of FOM compounds to cross BBB and determine its usefulness in the context of brain remyelination’.

Reviewer 2 Report

The manuscript contains interesting concepts. However, multiples pieces of experiments are still missing and the data are not sufficient to support the claims. I suggest the authors to make significant revisions before this manuscript could be published on Cells.

-        While the authors did identify the composition of fish oil, they did not isolate individual components but treated the hOPCs cells with whole oil. This begged the question: how could the authors separate the effects of nervonic acid from others?

-        Furthermore, there was no in vivo experiment where the diseased animals were treated with fish oil. Given that the model was available, this experiment is required to make any claims about the effects of fish oil.

-        The “dysregulated lipid composition” was identified in mouse model, but other experiments were performed on human cells. This created a discordance between the in vivo and in vitro data, raising the question: what are the evidences that the lipid compositions are the same in human and mice?

-        Figure 1 and table 2 are not enough to make such many claims regarding production and metabolisms of fatty acid. Please show the levels of intermediate metabolic products, or track the metabolism using radioactive carbons.

-        For the experiment assessing the effects of fish oil on production of cytokines and growth factor, the author only tested FGF2, IL-8 and IL-6, which seems to be random and incomplete. I suggest more complete testing using dot blots containing a wider selection of growth factors and cytokines.

-        The manuscript contains excessive abbreviation, making it very hard to read and follow. Some of the abbreviations were not accompanied by full term (such as HC, which I supposed to be heathy control).

-        Regarding the figures: In figure 3, what is the label and unit of graphs in panel A?

Author Response

1. We used the mixture of natural fish-derived oils to mimic the dietary conditions where a range of essential oils is consumed. The changes in lipid metabolism may depend on the environmental conditions and the availability of substrates. Moreover, multiple synergetic effects of fish-derived EPA and DHA were already documented, therefore we believe that NA separated effect would not be so beneficial and complete as an effect of a specially prepared formula of fish-derived oils. We speculate about NA essential role based on the three following aspects: 1. In the animal model, the main lipid dysregulation is observed at the level of NA in favor of AA synthesis; 2. supplementation of hOPCs by NA-rich oil results in its incorporation demonstrated by remarkable fluctuations in the lipid composition during hOPC maturation (24-72 h, corrected Table 2 and Figure 1C); 3. supplementation with LO with naturally lack of NA but possesses n-3 PUFAs  (supplementary Table 1) did not improve hOPCs functions. Appropriate changes have been made in the Discussion. ,One of the limitations in our study is the use of MS animal model to estimate fluctuations in the lipid composition, and subsequent speculation about inhibition of NA synthesis in MS  patients. Although EAE model accurately reflects the main pathological processes occurring during MS, human brain tissue has a significantly greater lipidome divergence compared to mice [52]. Another limitation is the use of fish-derived oil mixture instead of isolated NA. Natural FOM was chosen to mimic dietary conditions where a range of essential oils is consumed. The changes in the lipid metabolism may depend on the environmental conditions and the availability of substrates. As multiple synergetic effects of fish-derived EPA and DHA were already documented [9, 11, 15, 46-48], we speculate that NA separated function would not be so beneficial and complete as an effect of a specially prepared oil formula. Additionally, we failed to demonstrate any improvement of  hOPCs functions in the cultures supplemented with LO characterized by lack of NA but the presence of n-3 PUFAs.’

2.  In our opinion, the mouse model is suitable to observe general processes occurring in the brain. However, because of too much differences between human and mice brain tissue lipid composition, it is not reliable to make direct translation of what was found in the animal model into humans (Bozek, K., et al. Organization and evolution of brain lipidome revealed by large-scale analysis of human, chimpanzee, macaque, and mouse tissues. Neuron. 2015. 85, 695-702). Moreover, as tested FOM is of a natural fish-derived composition, we decided not to perform animal studies. The appropriate statement has been added in the Discussion (see below).

3. EAE is the most widely used animal model of MS. Recognizing the differences in the brain lipid composition between humans and mice, we believe that the majority of more universal processes is the same. We did not have a possibility to perform lipid analysis in the human MS tissue. At the same time, FOM supplementation is intendent to be studied in humans, therefore we employed hOPC model in this study. This limitation is described in the Discussion. The appropriate citation has been added.Bozek, K., et al. Organization and evolution of brain lipidome revealed by large-scale analysis of human, chimpanzee, macaque, and mouse tissues. Neuron. 2015. 85, 695-702

4. We have changed the Figure 1C to introduce products connected with NA synthesis. Moreover, the extended Table 2 presents the effect of FOM and LO on lipid composition in hOPCs. We also have added 17 additional products associated with metabolism of lipids as it was suggested. All intermediate products are presented in the Table 2. The new explanation has been added to the Figure 1 legend and the main text: ,Palmitic and stearic acid esters are constitutively presented in hOPCs and can be directly used for sphingomyelin synthesis or can also be used as substrates for NA synthesis with Cis-13-dicosenoic acid as an intermediate product (blue arrows). During FOM supplementation, NA is directly incorporated by hOPCs, while Cis-13-dicosenoic acid-dependent pathway is inhibited (red arrow). The rates of acid esters were measured at three time points (24h, 48 h and 72 h) in liophilisates of hOPCs incubated with FOM or LO or in medium’.

5. We have included additional results of 27 cytokines, chemokines and growth factors from multiplex assay (we have added Table 3 instead of Figure 4). The appropriated statistics and statements have been added.

,Therefore, using cytokine multiple profiling assay we have analyzed supernatant concentrations of twenty seven cytokines, chemokines and growth factors. We revealed that hOPCs cultured with FOM in comparison to hOPCs cultured in medium produced statistically significant increased amounts of fibroblast growth factor 2 (FGF-2) and vascular endothelial growth factor (VEGF), as well as decreased amounts of IL-5, IL-6, IL-7, IL-8, IL-15, IL-17, eotaxin, G-CSF, IFN-g, MCP-1 (Monocyte chemoattractant protein 1), and RANTES (Regulated on Activation, Normal T-cell Expressed and Secreted) (Table 3). Conversely, LO supplementation had no effect on IL-5, IL-6, IL-7, IL-8, IL-15, IL-17, eotaxin, FGF-2, MCP-1, RANTES and VEGF production by hOPC (Table 3). Taken together, FOM supplementation, in contrast to LO supplementation, tends to inhibit pro-inflammatory cytokine/chemokine production by hOPCs.’

6. We have checked and corrected the abbreviations and other language mistakes.

7. We have corrected the labelling in the Figure 3 in panel A.

Round 2

Reviewer 2 Report

The authors has addressed most of my concerns by either adding new data or rewriting the text appropriately. While I think the evidences are not very strong, I acknowledge that the study was hindered by technical limitations and not experimental designs. I recommend this manuscript to be published on Cells. However, please make sure that tables 2 and 3 will be printed correctly in the final version.